# Fruit and Vegetable Shopping Behavior and Intake among Low-Income Minority Households with Elementary-Aged Children

**DOI:** 10.3390/children10010082

**Published:** 2022-12-30

**Authors:** Brittni N. Metoyer, Ru-Jye Chuang, MinJae Lee, Christine Markham, Eric Brown, Maha Almohamad, Jayna M. Dave, Shreela V. Sharma

**Affiliations:** 1Department of Epidemiology, Human Genetics and Environmental Sciences, The University of Texas Health Science Center at Houston (UTHealth) School of Public Health, 1200 Pressler Street, Houston, TX 77030, USA; 2Department of Population and Data Sciences, Peter O’Donnell Jr. School of Public Health, University of Texas Southwestern Medical Center (UTSW), 5323 Harry Hines Blvd, Dallas, TX 75390, USA; 3Department of Health Promotion and Behavioral Sciences, The University of Texas Health Science Center at Houston (UTHealth) School of Public Health, 7000 Fannin Street, Houston, TX 77030, USA; 4USDA/ARS Children’s Nutrition Research Center, Department of Pediatrics, Baylor College of Medicine, 1100 Bates Ave, Houston, TX 77030, USA

**Keywords:** fruit and vegetable intake, low-income children, health promotion, shopping behavior, nutrition education, racial inequities

## Abstract

Low-income children and families do not meet the recommendations for fruit and vegetable (FV) intake. This study aimed to assess the association between FV shopping behavior and child FV intake through a cross-sectional study design analyzing self-reported surveys (*n* = 6074) from adult-child dyads of Hispanic/Latino and African American participants enrolled in the Brighter Bites co-op program. Through quantitative mixed effects linear regression models, accounting for school-level clustering and adjusting for covariates, child FV intake was positively associated with shopping for FV at large chain grocery stores (*p* < 0.001), natural/organic supermarkets (*p* < 0.001), warehouse club stores (*p* = 0.002), discount superstores (*p* < 0.001), small local stores/corner stores (*p* = 0.038), convenience stores (*p* = 0.022), ethnic markets (*p* = 0.002), farmers’ markets/co-op/school farm stands (*p* < 0.001), and gardens (*p* = 0.009) among Hispanic/Latinos participants. Among African American participants, there was significant positive association between child FV intake and shopping for FV at natural/organic supermarkets (*p* < 0.001), discount superstores (*p* = 0.005), and convenience stores (*p* = 0.031). The relationship between location and frequency of shopping for FV and child FV intake varied between races. Further research is needed to better understand the influence of cultural and physical environmental factors. Nutrition education programs are vital to encouraging families to make healthier food choices and purchases to improve child FV consumption.

## 1. Introduction

Childhood obesity has disproportionately affected low-income and minority children [1]. According to the 2017–2020 NHANES report, 26.2% of Hispanic children and 24.8% of African American children had obesity in the United States (U.S.) [1,2]. Children with obesity, particularly among low-income and minority families, have an increased risk for diet-related chronic conditions and diseases during childhood, with a greater likelihood of these tracking into adulthood, according to the latest CDC (Centers for Disease Control and Prevention) statistics on obesity [3,4,5,6]. Despite fruits and vegetables (FV) being one of the key components to a healthy diet with health-protective effects against diet-related chronic conditions like obesity and type 2 diabetes [3,7], children have not been meeting the recommended 2020- 2025 Dietary Guidelines for Americans for healthy eating patterns [3,8], especially low-income children who identify as African American or Hispanic [9,10,11].

Previous research showed low food access could further increase the risk of diet-related illnesses among low-income ethnically diverse families with significant barriers to healthful dietary behaviors [12]. Moreover, low-income and minority families were more likely to reside in neighborhoods that have greater access to fast-food restaurants, convenience stores, and other sources of food that promote unhealthy eating due to limited access of healthy food options [13]. In fact, many low-income and minority neighborhoods lack access to supermarkets and grocery stores; further limiting the access to FV [12,14,15,16,17]. Despite these findings, several studies have demonstrated that living near a supermarket is not correlated with more frequent store trips, food access, or FV intake [18,19,20]. Specifically, Caspi et al. [19], found FV consumption to be low among low-income families with adequate geographical access to supermarkets. In fact, many minority and low-income families travel outside of their neighborhood to shop for groceries, especially African American families [18,20].

Previous studies have found a positive association between individuals who shop at supermarkets, discount superstores, and farmers’ markets and FV intake among adults [21,22,23,24,25]. Moreover, individuals who shop at convenience stores reported more frequent purchases of less healthy foods such as junk food and lower FV consumption as compared to individuals who shopped at supermarkets and non-convenience stores, respectively [23,24,25]. A study also found that Hispanics purchase more FV than African Americans [26]. Currently, there is limited research examining where and how frequent low-income families shop and purchase FV from various types of food stores, and their associations with child FV intake. Thus, this study fills a gap in existing literature by examining this relationship.

The purpose of this study is to examine the association between FV shopping behaviors and child FV intake among elementary-aged Hispanic/Latino and African American children participating in the Brighter Bites school-based food co-op program in the 2018–2019 school year. Brighter Bites is an evidence-based health promotion program that increases access to FV and nutrition education to improve FV intake and the home nutrition environment among low-income families [27,28,29].

## 2. Materials and Methods

### 2.1. Study Design

This study was a secondary analysis of cross-sectional baseline parent survey data from Hispanic/Latino and African American participating families. Surveys were collected in the fall semester before the 2018–2019 Brighter Bites program began for the school year.

### 2.2. Brighter Bites

Brighter Bites is a theoretically grounded, evidence-based health promotion program currently implemented in 11 cities across the United States. In the 2018–2019 school year, Brighter Bites was implemented in only 6 cities, Houston, Dallas, Austin, New York City, Washington D.C., and rural area of Southwest Florida. The goal of this nonprofit is to increase access to FV and nutrition education, as an approach to improve FV consumption and the home nutrition environment among low-income children and families [27,28,29]. This 16-week school-based food co-op program consists of weekly distributions of fresh produce (~50–60 servings per family), implementation of evidence-based nutrition education, validated Coordinated Approach to Child Health (CATCH) curriculum focusing on healthy nutrition and physical activity [30], sent-home bilingual nutrition education materials and recipe cards, and weekly recipe tastings held at the distribution sites [29]. A comprehensive explanation of the description and methodology of Brighter Bites has been provided elsewhere [27]. Prior evidence showed Brighter Bites significantly improved FV intake and the home nutrition environment among participating families [27,28,29,31].

### 2.3. Participants

In order for a public or charter elementary school to be eligible to enroll in Brighter Bites, they would have to meet the following criteria: either (1) at least 80% of its student population would be eligible for free or reduced-price lunch, or (2) received Title I funding, and a commitment to implementation of the CATCH curriculum in classrooms. After an eligible school has elected and consented to participate in the program, participants were recruited in parent–child dyads, where parent is defined as any primary caregiver in the household. Recruitment consisted of informational fliers sent home to parents and distributed at the back-to-school parent meetings. All children from eligible schools were eligible to participate in Brighter Bites. Informed consent was obtained when parents enrolled their family into the program.

### 2.4. Data Collection

Self-report surveys were administered in paper or digital format sent via text to the enrolled families, in English and Spanish, to the primary adult caregiver of all the children in the Brighter Bites participating family at the time of enrollment until week two of program implementation for the 2018–2019 school year programming. Completion of the self-report survey was voluntary. Overall, of the 23,694 families enrolled in Brighter Bites in the 2018–2019 school year, a convenience sample of 6074 surveys were collected from Hispanic/Latino and African American participating households in September-October 2018 and analyzed as a part of this study (26% response rate). At least one survey was collected from each of the 87 participating schools across the five cities: Houston, Dallas, Austin, Washington D.C., and Southwest Florida in the 2018–2019 year of Brighter Bites implementation, except for New York City due to city-specific regulations. All data were collected by Brighter Bites as a part of the ongoing program evaluation efforts, and de-identified data were shared with University of Texas Health Science Center (UTHealth) for analysis. This study was approved by UTHealth, Committee for Protection of Human Subjects.

### 2.5. Survey Measures

#### 2.5.1. Child FV Intake

Child FV intake was measured using questions adapted from the previously validated 27-item National Cancer Institute’s 2014 Family Life, Activity, Sun, Health, and Eating Study (FLASHE) dietary screener [32]. Parents were asked how many times in the past week did their child drink 100% pure fruit juice and eat fruit, green salad or non-fried vegetables, and fried and other kinds of potatoes. A total of seven questions were used to assess FV intake. The response options included: never, 1–2 times per week, 3–4 times per week, 5–6 times per week, and 7 times per week. Responses from each question were further categorized into a composite outcome variable “child FV intake”, with a continuous scale ranging from 0–5 times per day.

#### 2.5.2. FV Shopping Behavior

FV shopping behavior was assessed using questions adapted from the previously validated National Cancer Institute’s 2007 Food Attitudes and Behaviors (FAB) Survey screener [33]. Parents were asked to report how often did they buy or obtain FV for the family from the following stores: large chain grocery stores (e.g., Albertsons, H-E-B, Kroger), natural or organic supermarkets (e.g., Whole Foods or Sprouts), warehouse club stores (e.g., Sam’s Club or Costco), discount superstores (e.g., Wal-Mart or Target), small local stores or corner stores (e.g., usually locally owned and do not sell gas), convenience stores (e.g., 7–11 or mini-market, usually sells gas), ethnic markets (e.g., Asian, Indian, or Hispanic), farmers’ markets/co-ops/school farm stands, food banks/pantries, and personal gardens. One question was used to assess shopping behavior. One example of each store type was provided on the survey, except for farmers’ markets and food banks. The response options were as follows: never, less than once a month, 1–2 times per month, 1 time per week, and 2+ times per week.

#### 2.5.3. Demographic Characteristics

Demographics included parent race/ethnicity, parent age, gender and age of the parent’s youngest child, respondents’ relationship to child, language spoken at home, household size, parent education level, parent employment status, and enrollment in government assistance programs [Special Supplemental Nutrition Program for Women, Infants, and Children (WIC), Supplemental Nutrition Assistance Program (SNAP), Double Dollars, Medicaid, Medicare, National School Lunch Program and/or School Breakfast Programs (NSLP/SBP), and Children’s Health Insurance Program (CHIP)].

#### 2.5.4. Household Food Security Status

Household food security status was measured using the validated 2-item Hunger Vital Sign screener [34]. Participants were considered food insecure if they responded, often true or sometimes true to either “Within the past 3 months we worried whether our food would run out before we got money to buy more.” or “Within the past 3 months the food we bought just didn’t last and we didn’t have money to get more.”

### 2.6. Data Analysis

All analyses were performed using STATA 15.1 (StataCorp, College Station, TX, USA). Frequencies, means, and standard deviations (SD) were computed for all variables. Pearson’s Chi-square tests for categorical variables and t-tests for continuous variables were used to test differences between variables by race/ethnicity for descriptive purposes. Significance was denoted by *p* < 0.05 and 95% confidence interval (CI). To account for school-level clustering, mixed effects linear regression models with participants (level 1) nested within school site (level 2) were used to assess the cross-sectional association between the frequency of shopping for fruits and vegetables (exposure) and child FV intake (outcome) within each racial/ethnic group at baseline prior to participating in Brighter Bites. Purposeful selection model building with significance *p* < 0.10 forward and *p* < 0.05 backward and change in estimate method (>10%) identified significant demographic confounding variables that were adjusted for in each model.

## 3. Results

### 3.1. Parent Demographic Characteristics

The demographic characteristics of survey respondents enrolled in Brighter Bites for the 2018–2019 school year (*n* = 6074) are presented in Table 1. A total of 5601 (92.2%) respondents identified as Mexican American, Hispanic, or Latino (hereafter referred to as “Hispanic”) and 473 (7.8%) respondents identified as African American or Black (hereafter referred to as “African American”). Adult respondents were on an average 34.8 years old (±7.8); majority were mothers (93%), located in Houston, Texas (51.8%), homemakers (53.7%), received no college education (75.8%), and were newly participating in the Brighter Bites program this school year (57.7%, data not shown in table). The average age of children from participating households was 6.5 years old (±2.0). The NSLP, Medicaid, and SNAP were the most utilized government assistance programs (59.9%, 74.8%, and 34.2%; respectively) and nearly three-fourths (71.8%) of respondents reported being food insecure. When examining demographic differences by race/ethnicity, compared to Hispanic respondents, those who identified as African American had smaller number of persons in their households (4.7 vs. 5.1; *p* < 0.001), were employed full-time or part-time (61.7% vs. 28.9%; *p* < 0.001) and had slightly more households experiencing food insecurity (76.0% vs. 71.5%, *p* = 0.046), and were more likely to participate in SNAP (48.2% vs. 33.0%, *p* < 0.001) and NSLP (84.5% vs. 74.0%, *p* < 0.001).

### 3.2. Child FV Intake

Overall, prior to participating in Brighter Bites (i.e., at baseline), children frequency of FV consumption across all respondents was 2.04 times per day. Children of Hispanic parent respondents were reportedly eating FV 2.02 times per day, while children of African American parents were eating FV 2.34 times per day (*p* < 0.001). [data are not presented in a table].

### 3.3. FV Shopping Behaviors

The description of FV shopping behaviors at baseline by race/ethnicity is presented in Table 2. Majority of respondents reportedly shopped for their FV at the large chain grocery stores (66.7% shopped 1+ times/week), followed by discount superstores (32.1% shopped 1–2 times/month), and warehouse club stores (24.4% shopped 1–2 times/month). A significant proportion of respondents reportedly never shopped at natural or organic supermarkets (69.8%), small local stores or corner stores (59.4%), convenience stores (74.1%), ethnic markets (54.8%), farmers’ market/co-ops/school farm stands (83.3%), food banks/pantries (83.1%), and gardens (92.4%) for FV. Significant differences were found in the frequency of FV shopping between Hispanic and African American respondents (*p* < 0.05). Compared to Hispanics, African Americans shopped for FV more frequently at large chain grocery stores (37.6% shopped 1+ times/week vs. 25.0%, *p* < 0.001), discount superstores (24.0% shopped 1+ times/week vs. 32.4%, *p* < 0.001) and warehouse club stores (14.0% shopped 1+ times/week vs. 9.0%, *p* < 0.001).

### 3.4. Relationship between FV Shopping and Child FV Intake

After adjusting for covariates, results of regression analysis showed that, among Hispanics there was a significant positive association between child FV intake and shopping for FV at large chain grocery stores (*p* < 0.001), natural or organic supermarkets (*p* < 0.001), warehouse club stores (*p* = 0.002), discount superstores (*p* < 0.001), small local stores or corner stores (*p* = 0.038), convenience stores (*p* = 0.022), ethnic markets (*p* = 0.002), farmers’ markets/co-op/school farm stands (*p* < 0.001), gardens (*p* = 0.009) (Table 3). Covariates that were adjusted for include: parent age, child age, household size, city, child gender, parental employment, language spoken at home, parental education, food security status, and government assistance programs (WIC, SNAP, Double Dollars, Medicaid, Medicare, Free and reduced meals/NSLP, CHIP). However, each model had different significant covariates. In addition, a dose–response relationship was observed between large chain grocery stores, natural or organic supermarkets, warehouse club stores, discount superstores, and food bank/pantry and child FV intake. Among African Americans, there was also a significant positive association between child FV intake and frequency of shopping at natural or organic supermarkets (*p* < 0.001), discount superstores (*p* = 0.005), and convenience stores (*p* = 0.031), after adjusting for covariates in each model. The same dose–response relationship was only found between natural or organic supermarkets and warehouse club stores and child FV intake.

## 4. Discussion

Our study identified the type of stores and frequency of FV shopping among participating households, examined the cross-sectional association between frequency of FV shopping and child FV intake among low-income minority families participating in the Brighter Bites health promotion intervention. To the best of our knowledge, this cross-sectional study is the first to examine this association in low-income school-aged predominantly Hispanic/Latino and African American households with children who are also experiencing high rates of food insecurity.

Most respondents reported shopping more often at large chain stores, discount superstores, and warehouse clubs for FV than small local stores or corner stores and convenience stores. Our findings concur with the current literature that show African Americans and Hispanics primarily shop at larger grocery stores and superstores [21,22,24,25,35]. While previous studies have found minority and low-income populations to shop more frequently at supermarkets or supercenters [21,24,25]; some studies have shown these populations to have significantly less access to larger grocery stores and supermarkets [12,14,15,16,17]. Our results indicated that regardless of where the respondents reside, majority of them obtain their FV from large grocery stores or supermarkets, while few shopped at local convenience stores or corner stores. Thus, proximity and distance may be less influential in their shopping decision as compared to other considerations such as convenience of shopping for all foods and other non-food items at one location, or variety, quality and pricing of produce available at larger stores [12,20,36,37].

In this study, 71.9% of families were food insecure, 34.3% of families participated in SNAP and 26.8% participated in WIC. Where respondents shop for produce may be influenced by their food security status, the number of benefits received each cycle, timing of the benefits, and where benefits are accepted [38]. According to 2019 U.S. Department of Agriculture (USDA) Food and Nutrition Service Benefit Redemption Report, SNAP participants redeemed over 82% of their benefits at supermarkets and superstores; however, these stores made-up 15% of the total number of SNAP-authorized stores in the U.S that year [39]. WIC benefits are also primarily redeemed at large chain stores (85%) [39]. Government nutrition assistance programs were designed to improve access to nutritious foods by providing low-income families with benefits to purchase healthful foods. Furthermore, in this study, only a small proportion of respondents shopped for FV at natural or organic supermarkets, ethnic markets, farmers’ markets/co-ops/school farm stands, food banks/pantries, and gardens. This finding aligns with findings from a previous qualitative study that found farmers’ markets, community gardens, and stands were less visited among African American and Latino mothers [35]. In the same study, African American mothers were less open to shopping at farmers’ markets for produce than Latino mothers [35], but these results were not consistent with our findings that found African American respondents shopped more often at farmers’ markets than Hispanic respondents Prior studies have shown that factors that limit shopping in natural supermarkets and farmers’ markets include perceived high costs of produce, lack of awareness and availability in neighborhoods, safety of farming practices, does not accept SNAP benefits, and variety, quality and pricing of produce [25,35,40,41,42]. Additionally, studies have found cultural preferences for produce and other grocery food items to significantly impact where minority populations shop [25,35]. These results, along with those in our study which included data across five cities in the U.S., indicate the need for accessible grocery outlets where families can shop for all their food needs, including fruits and vegetables, at affordable prices, consider local cultures and preferences, and participate in government assistance programs such as WIC and SNAP.

Among Hispanic/Latinos, a positive association between frequency of FV shopping and child frequency of FV intake was found in all but one food store type (food bank/pantry). Similarly, the same positive association was found in natural/organic supermarkets, superstores, and convenience stores among African Americans. Our results also demonstrated that the magnitude of association was stronger with increased frequency of visits to certain food stores, and the type of food store differed among Hispanic/Latino and African American shoppers. While it is unclear from our study whether child demand for FV preceded the grocery shopping behavior or vice versa, these results provide preliminary evidence that frequency of grocery shopping for FV and child frequency of consumption of FV are positively associated, indicating that access to retail outlets, especially large grocery stores and supermarkets, is of relevance to strategies aiming to increase child consumption of FV among low-income families. Our findings also indicated that shopping at certain types of food stores for produce may be associated with FV intake of African American and Hispanic/Latino children differently. Proximity and access to convenience stores has been associated with lower fruit consumption [43], unhealthy eating behaviors [44], and weight gain [45] in African American children. One study found the odds of FV purchases at convenience stores to decrease as the variety of FV available to purchase decreased [46]. This study also found that if convenience stores placed FV in front aisles, purchases of sugar-sweetened beverages decreased among Hispanics and African Americans [46]. More research is needed to better understand the ethnic, cultural and physical environment factors that may influence the relationship between shopping for FV and child FV consumption. Future nutrition programs and in-store grocery interventions, after further research is conducted, may need to be culturally tailored to the target racial and ethnic group.

### 4.1. Strengths

One strength of this study included the use of an extensive list of food store types to measure FV shopping behavior. In addition, the relationship between food shopping behaviors and child FV intake evaluated among Hispanic/Latino and African American populations independently, as opposed to one low-income minority group. Lastly, this study included a large sample size of low-income, minority individuals from various geographic areas of the US (Houston, Austin, Dallas, Washington DC and SW Florida).

### 4.2. Limitations

One limitation was that this study used cross-sectional data, thus causality cannot be inferred. Second, our survey asked respondents about their food shopping behaviors for FV only, so we were unable to make direct comparisons to previous studies that have examined shopping behaviors for all grocery food items. Third, determinants of shopping behaviors (e.g., proximity, distance, cost) were not assessed, as they were beyond the scope of the study. Future research examining important predictors of produce purchasing may provide insight to our findings. Fourth, frequency and location of shopping and dietary intake measures were self-reported; therefore, there is potential for social desirability bias. Last, this study used a convenience sample from a low-income minority population, which could limit the generalizability of study findings. Self-selection bias may be present as more Hispanic/Latino families (*n* = 5601) enrolled in Brighter Bites and responded to the survey than African American families (*n* = 473). Nevertheless, our study emphasizes the importance of evaluating FV shopping behaviors of parents to promote FV consumption of children.

## 5. Conclusions

The results of this study support existing literature that purchase and consumption of produce are associated with each other. The place and frequency of purchasing FV are important behavioral aspects that define the nutritional quality of these health necessities. This study found most low-income minority families with children shop more frequently at large chain stores, superstores, and warehouse clubs for produce than convenience stores, local/corner stores, natural/organic supermarkets, and farmers’ markets/co-ops. Additionally, shopping more often for FV was positively associated with child FV intake. However, this association differed by food store type and race/ethnicity. More research is needed to identify the influential factors in FV shopping decisions and determine how these factors may vary by race and ethnicity. Notably, education and motivating families to make healthier food choices no matter where they choose to shop is essential to improve child FV consumption. Future programs and in-store grocery interventions may need to be tailored to address influences that match the child and families’ race, culture, ethnicity and physical environment. Finally, the COVID-19 pandemic may have impacted family purchasing and consumption behaviors especially among low-income households with children. Future studies understanding changes in these behaviors as a result of the pandemic may be warranted.

## Figures and Tables

**Table 1 children-10-00082-t001:** Demographic Characteristics of Brighter Bites Participants (*n* = 6074) Stratified by Race/Ethnicity, Fall 2018 Parent Survey at baseline.

Variable	Total (*n* = 6074)	Hispanic/Latino (*n* = 5601)	African American (*n* = 473)	*p* Value *
	*mean* ± *sd*			
Age (years)				
Adult	34.87 ± 7.787	34.57 ± 7.44	36.92 ± 9.24	**<0.001**
Child	6.53 ± 2.00	6.53 ± 1.99	6.34 ± 2.10	0.096
Household size (*n*)				
Adult	2.45 ± 1.09	2.48 ± 1.09	2.15 ± 1.14	**<0.001**
Child	2.62 ± 1.17	2.64 ± 1.16	2.60 ± 1.26	0.529
Total	5.05 ± 1.58	5.08 ± 1.56	4.65 ± 1.73	**<0.001**
	*n (col %)*	*n (row %)*		
City				**<0.001**
Austin	871 (14.34)	787 (90.36)	84 (9.64)	
Dallas	1572 (25.86)	1479 (94.14)	92 (5.86)	
Houston	3145 (51.78)	2940 (93.48)	205 (6.52)	
SW Florida	231 (3.80)	221 (95.67)	10 (4.33)	
Washington DC	256 (4.21)	174 (67.97)	82 (32.03)	
		*n (col %)*		
Child Gender				0.122
Male	3019 (50.74)	2802 (51.03)	217 (47.28)	
Female	2931 (49.26)	2689 (48.97)	242 (52.72)	
Parental employment				**<0.001**
Employed (full/part time)	1721 (31.39)	1460 (28.86)	261 (61.70)	
Self-employed	312 (5.69)	282 (5.57)	30 (7.09)	
Homemaker	2944 (53.70)	2892 (57.17)	52 (12.29)	
Unemployed	400 (7.30)	347 (6.86)	53 (12.53)	
Unable to work	105 (1.92)	78 (1.54)	27 (6.38)	
Language spoke at home				**<0.001**
English only	968 (16.08)	546 (9.84)	422 (89.79)	
Bilingual	2157 (35.82)	2150 (38.73)	7 (1.49)	
Spanish only	2852 (47.35)	2845 (51.25)	6 (1.28)	
Other language(s)	45 (0.75)	10 (0.18)	35 (7.45)	
Parental education				**<0.001**
Never attended school or only kindergarten	77 (1.38)	77 (1.49)	0	
Grades 1–11	1015 (18.13)	1009 (19.57)	6 (1.35)	
Grades 9–11	1172 (20.93)	1145 (22.21)	27 (6.08)	
Grade 12 or GED	1981 (35.38)	1830 (35.50)	151 (34.01)	
College 1–3 years	1005 (17.95)	851 (16.51)	154 (34.68)	
College graduate or more	349 (6.23)	243 (4.71)	106 (23.87)	
Food Security Status				**0.046**
Food insecure	4075 (71.88)	3734 (71.53)	341 (75.95)	
Food secure	1594 (28.12)	1486 (28.47)	108 (24.05)	
	*n (yes %)*			
Government Assistance Program				
WIC	1581 (26.84)	1470 (27.03)	111 (24.50)	0.243
SNAP	2001 (34.23)	1780 (33.04)	221 (48.15)	**<0.001**
Double Dollars	63 (1.09)	52 (0.97)	11 (2.46)	**0.004**
Medicaid	3488 (59.59)	3243 (60.03)	245 (54.32)	**0.018**
Medicare	354 (6.10)	277 (5.17)	77 (17.34)	**<0.001**
Free + reduced meals/NSLP	4315 (74.84)	3928 (74.00)	387 (84.50)	**<0.001**
CHIP	1204 (20.96)	1102 (20.78)	102 (23.08)	0.255

Abbreviations: CHIP, Children’s Health Insurance Program; SNAP, Supplemental Nutrition Assistance Program; WIC, Special Supplemental Nutrition Program for Women, Infants, and Children; GED, General Educational Development test; NSLP, National School Lunch Program. * Boldface denotes significant differences (*p* < 0.05) between ethnic groups; determined by t-test or chi-square test.

**Table 2 children-10-00082-t002:** Reported Fruit and Vegetable (FV) Food Shopping Behaviors of Brighter Bites Participants (*n* = 6074) Stratified by Race/Ethnicity, Fall 2018 Parent baseline survey.

Store Type and Frequency of Shopping	Total (*n* = 6074)	Hispanic/Latino (*n* = 5601)	African American (*n* = 473)	*p* Value *
	*n (col %)*			
Large chain grocery store ^a^				**<0.001**
Never	438 (7.43)	413 (7.61)	25 (5.36)	
Less than once a month	435 (7.38)	400 (7.37)	35 (7.51)	
1–2 times per month	1088 (18.45)	952 (17.53)	136 (29.18)	
1 time per week	2404 (40.77)	2309 (42.52)	95 (20.39)	
2+ times per week	1531 (25.97)	1356 (24.97)	175 (37.55)	
Natural or organic supermarket ^b^				**<0.001**
Never	3988 (68.79)	3743 (70.11)	245 (53.49)	
Less than once a month	761 (13.13)	670 (12.55)	91 (19.87)	
1–2 times per month	515 (8.88)	459 (8.60)	56 (12.23)	
1 time per week	365 (6.30)	328 (8.08)	37 (9.75)	
2+ times per week	168 (2.90)	139 (2.69)	29 (6.33)	
Warehouse club store ^c^				**<0.001**
Never	2375 (40.84)	2216 (41.35)	159 (34.79)	
Less than once a month	1475 (25.36)	1373 (25.62)	102 (22.32)	
1–2 times per month	1420 (24.42)	1288 (24.03)	132 (28.88)	
1 time per week	346 (5.95)	319 (5.95)	27 (5.91)	
2+ times per week	200 (3.44)	163 (3.04)	37 (8.10)	
Discount superstore ^d^				**<0.001**
Never	754 (12.83)	700 (12.92)	54 (11.69)	
Less than once a month	1309 (22.27)	1233 (22.76)	76 (16.45)	
1–2 times per month	1886 (32.08)	1729 (31.92)	157 (33.98)	
1 time per week	1136 (19.32)	1072 (19.79)	64 (13.85)	
2+ times per week	794 (13.51)	683 (12.61)	111 (24.03)	
Small local store or corner store ^e^				**0.043**
Never	3446 (59.41)	3171 (59.38)	275 (59.78)	
Less than once a month	843 (14.53)	782 (14.64)	61 (13.26)	
1–2 times per month	660 (11.38)	595 (11.14)	65 (14.13)	
1 time per week	525 (9.05)	497 (9.31)	28 (6.09)	
2+ times per week	326 (5.62)	295 (5.52)	31 (6.74)	
Convenience store ^f^				**<0.001**
Never	4271 (74.06)	3973 (74.86)	298 (64.78)	
Less than once a month	779 (13.51)	707 (13.32)	72 (15.65)	
1–2 times per month	353 (6.12)	311 (5.86)	42 (9.13)	
1 time per week	240 (4.16)	220 (4.15)	20 (4.35)	
2+ times per week	124 (2.15)	96 (1.81)	28 (6.09)	
Ethnic market ^g^				**<0.001**
Never	3162 (54.76)	2855 (53.71)	307 (67.03)	
Less than once a month	832 (14.41)	761 (14.32)	71 (15.50)	
1–2 times per month	749 (12.97)	702 (13.21)	47 (10.26)	
1 time per week	696 (12.05)	681 (12.81)	15 (3.28)	
2+ times per week	335 (5.80)	317 (5.96)	18 (3.93)	
Farmers’ market/co-op/school farm stand				**<0.001**
Never	4782 (83.28)	4513 (85.36)	269 (59.12)	
Less than once a month	514 (8.95)	426 (8.06)	88 (19.34)	
1–2 times per month	265 (4.62)	209 (3.95)	56 (12.31)	
1 time per week	121 (2.11)	99 (1.87)	22 (4.84)	
2+ times per week	60 (1.04)	40 (0.76)	20 (4.40)	
Food bank/pantry				**<0.001**
Never	4767 (83.11)	4433 (83.91)	334 (73.73)	
Less than once a month	507 (8.84)	445 (8.42)	62 (13.69)	
1–2 times per month	300 (5.23)	260 (4.92)	40 (8.83)	
1 time per week	119 (2.07)	110 (2.08)	9 (1.99)	
2+ times per week	43 (0.75)	35 (0.66)	8 (1.77)	
Garden				**0.015**
Never	5340 (92.39)	4934 (92.66)	406 (89.23)	
Less than once a month	202 (3.49)	184 (3.46)	18 (3.96)	
1–2 times per month	121 (2.09)	104 (1.95)	17 (3.74)	
1 time per week	69 (1.19)	63 (1.18)	6 (1.32)	
2+ times per week	48 (0.83)	40 (0.75)	8 (1.76)	

* Boldface denotes significant differences (*p* < 0.05) between ethnic groups; determined by t-test or chi-square test. ^a^ e.g., Randall’s, HEB, Kroger’s, Fiesta. ^b^ e.g., Whole Foods or Sprouts. ^c^ e.g., Sam’s Club or Costco. ^d^ e.g., Wal-Mart or Target. ^e^ e.g., Usually locally owned and do not sell gas. ^f^ e.g., 7–11 or mini market, usually sell gas. ^g^ e.g., Asian, Indian, or Hispanic.

**Table 3 children-10-00082-t003:** Association Between Fruit and Vegetable (FV) Food Shopping Patterns and Child FV intake, Stratified by Race/Ethnicity.

Store Type and Frequency of Shopping	Hispanic/Latino		African American	
	Mean Child FV Intake ß ^§^ (95% CI)	*p* Value *	Mean Child FV Intake ß ^§^ (95% CI)	*p* Value *
Large chain grocery store ^a^ (reference: never)				
Less than once a month	0.041 (−0.090, 0.171) ^1^	0.542	0.105 (−0.378, 0.589) ^7^	0.669
1–2 times per month	0.110 (−0.001, 0.222) ^1^	0.053	−0.103 (−0.521, 0.315) ^7^	0.630
1 time per week	0.193 (0.090, 0.297) ^1^	**<0.001**	0.085 (−0.351, 0.522) ^7^	0.701
2+ times per week	0.310 (0.201, 0.419) ^1^	**<0.001**	0.122 (−0.298, 0.541) ^7^	0.570
Natural or organic supermarket ^b^ (reference: never)				
Less than once a month	0.124 (0.050, 0.198) ^1^	**0.001**	0.068 (−0.145, 0.280) ^7^	0.532
1–2 times per month	0.177 (0.089, 0.265) ^1^	**<0.001**	0.179 (−0.091, 0.448) ^7^	0.194
1 time per week	0.312 (0.213, 0.412) ^1^	**<0.001**	0.480 (0.153, 0.807) ^7^	**0.004**
2+ times per week	0.393 (0.243, 0.543) ^1^	**<0.001**	0.912(0.548, 1.28) ^7^	**<0.001**
Warehouse club store ^c^ (reference: never)				
Less than once a month	0.022 (−0.036, 0.081) ^2^	0.457	0.070 (−0.169, 0.308) ^8^	0.568
1–2 times per month	0.110 (0.049, 0.171) ^2^	**<0.001**	0.076 (−0.152, 0.303) ^8^	0.515
1 time per week	0.156 (0.053, 0.260) ^2^	**0.003**	0.180 (−0.244, 0.604) ^8^	0.406
2+ times per week	0.217 (0.080, 0.353) ^2^	**0.002**	0.323 (−0.031, 0.677) ^8^	0.074
Discount superstore ^d^ (reference: never)				
Less than once a month	0.089 (0.005, 0.172) ^3^	**0.037**	0.214 (−0.102, 0.531) ^9^	0.184
1–2 times per month	0.126 (0.048, 0.205) ^3^	**0.002**	0.233 (−0.050, 0.517) ^9^	0.107
1 time per week	0.180 (0.095, 0.266) ^3^	**<0.001**	0.171 (−0.154, 0.497) ^9^	0.302
2+ times per week	0.205 (0.112, 0.299) ^3^	**<0.001**	0.424 (0.129, 0.719) ^9^	**0.005**
Small local store or corner store ^e^ (reference: never)				
Less than once a month	0.092 (0.021, 0.162) ^1^	**0.011**	−0.023 (−0.265, 0.219) ^10^	0.853
1–2 times per month	0.010 (−0.069, 0.090) ^1^	0.798	0.182 (−0.064, 0.429) ^10^	0.147
1 time per week	0.039 (−0.046, 0.123) ^1^	0.367	0.048 (−0.308, 0.403) ^10^	0.793
2+ times per week	0.115 (0.006, 0.223) ^1^	**0.038**	0.339 (−0.007, 0.683) ^10^	0.055
Convenience store ^f^ (reference: never)				
Less than once a month	0.072 (−0.000, 0.144) ^4^	0.050	0.050 (−0.179, 0.278) ^11^	0.671
1–2 times per month	0.065 (−0.042, 0.172) ^4^	0.233	0.313 (0.021, 0.604) ^11^	**0.035**
1 time per week	0.117 (−0.004, 0.238) ^4^	0.059	0.904 (0.506, 1.30) ^11^	**<0.001**
2+ times per week	0.219 (0.032, 0.406) ^4^	**0.022**	0.387 (0.035, 0.738) ^11^	**0.031**
Ethnic market (reference: never)				
Less than once a month	0.032 (−0.038, 0.102) ^5^	0.369	0.285 (0.019, 0.552) ^12^	**0.036**
1–2 times per month	0.028 (−0.045, 0.101) ^5^	0.454	0.163 (−0.175, 0.501) ^12^	0.344
1 time per week	0.084 (0.012, 0.157) ^5^	**0.023**	0.045 (−0.517, 0.608) ^12^	0.874
2+ times per week	0.160 (0.058, 0.261) ^5^	**0.002**	−0.054 (−0.590, 0.481) ^12^	0.843
Farmers’ market/co-op/school farm stand (reference: never)				
Less than once a month	0.151 (0.060, 0.242) ^6^	**0.001**	0.131 (−0.099, 0.361) ^13^	0.265
1–2 times per month	0.272 (0.142, 0.401) ^6^	**<0.001**	0.296 (0.018, 0.574) ^13^	**0.037**
1 time per week	0.217 (0.027, 0.408) ^6^	**0.025**	0.424 (0.027, 0.821) ^13^	**0.036**
2+ times per week	0.631 (0.309, 0.954) ^6^	**<0.001**	−0.049 (−0.478, 0.381) ^13^	0.824
Food bank/pantry (reference: never)				
Less than once a month	0.051 (−0.039, 0.141) ^1^	0.269	0.137 (−0.115, 0.389) ^14^	0.285
1–2 times per month	0.116 (−0.000, 0.232) ^1^	0.052	0.236 (−0.066, 0.539) ^14^	0.125
1 time per week	0.129 (−0.044, 0.303) ^1^	0.144	0.091 (−0.555, 0.738) ^14^	0.782
2+ times per week	0.184 (−0.127, 0.495) ^1^	0.247	0.643 (−0.220, 1.51) ^14^	0.144
Garden (reference: never)				
Less than once a month	0.116 (−0.020, 0.251) ^4^	0.094	0.288 (−0.277, 0.853) ^15^	0.317
1–2 times per month	0.013 (−0.170, 0.196) ^4^	0.889	−0.133 (−0.718, 0.451) ^15^	0.654
1 time per week	0.187 (−0.054, 0.429) ^4^	0.129	0.461 (−0.668, 1.59) ^15^	0.424
2+ times per week	0.383 (0.096, 0.669) ^4^	**0.009**	0.218 (−0.605, 1.04) ^15^	0.604

^§^ Regression coefficients were calculated using Multilevel Mixed Effects Linear Regression models. * Boldface indicates statistical significance at *p* < 0.05. ^a^ e.g., Randall’s, HEB, Kroger’s Fiesta. ^b^ e.g., Whole Foods or Sprouts. ^c^ e.g., Sam’s Club or Costco. ^d^ e.g., Wal-Mart or Target. ^e^ e.g., Usually locally owned and do not sell gas. ^f^ e.g., 7–11 or mini market, usually sell gas. ^g^ e.g., Asian, Indian, or Hispanic. ^1^ Adjusted for child race, language spoken at home, free and reduced meals, Medicare, SNAP, parent age, child age, city, and food insecurity status. ^2^ Adjusted for language spoken at home, free and reduced meals, Medicare, SNAP, child age, and food insecurity status. ^3^ Adjusted for child race, language spoken at home, free and reduced meals, Medicare, SNAP, child age, and food insecurity status. ^4^ Adjusted for child race, language spoken at home, free and reduced meals, Medicare, SNAP, parent age, child age, and food insecurity status. ^5^ Adjusted for child race, language spoken at home, Medicare, SNAP, child age, and food insecurity status. ^6^ Adjusted for child race, language spoken at home, free and reduced meals, Medicare, parent age, child age, city, and food insecurity status. ^7^ Adjusted for food insecurity status, free and reduced meals, SNAP, parent employment, parent education, child grade, and city. ^8^ Adjusted for food insecurity status, free and reduced meals, SNAP, parent employment, child grade, city, double dollars, language spoken at home, total household size, child race, CHIP, Medicare, and child gender. ^9^ Adjusted for food insecurity status, free and reduced meals, SNAP, double dollars, and city. ^10^ Adjusted for food insecurity status, free and reduced meals, SNAP, double dollars, CHIP, child grade, and city. ^11^ Adjusted for food insecurity status, free and reduced meals, SNAP, CHIP, and child grade. ^12^ Adjusted for food insecurity status, free and reduced meals, SNAP, parent employment, parent education, child grade, city, double dollars, language spoken at home, child race, and parent age. ^13^ Adjusted for food insecurity status, free and reduced meals, SNAP, parent employment, WIC, child grade, and city. ^14^ Adjusted for food insecurity status, free and reduced meals, SNAP, parent employment, double dollars, child grade, child gender, and city. ^15^ Adjusted for food insecurity status, free and reduced meals, SNAP, parent employment, child grade, parent education, city, double dollars, language spoken at home, WIC, child race, CHIP, Medicare, and parent age.

## Data Availability

The data presented in this study are available on request from the corresponding author. The data are not publicly available due to restrictions: Privacy or Ethical.

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
