# Peer review of "Fruit and Vegetable Shopping Behavior and Intake among Low-Income Minority Households with Elementary-Aged Children"

_children, 2022, doi:10.3390/children10010082_

Round 1

Reviewer 1 Report

Dear authors, thank you for your contribution. In a world where childhood obesity largely affects children from minority and low-income families, the presented study is very useful.

The consumption of fruits and vegetables is very important for children's health, nutritional education having an essential role in the purchase and consumption of these categories of products.

The place and frequency of purchase of fruits and vegetables are important behavioral aspects that define the nutritional quality and freshness of products.

The structure of the article is logical and the presentation of the results is clear.

Unfortunately, one feels the lack of an important chapter of the work, namely Conclusions.

Chapter 5. Implications for Research and Practice is very important but cannot replace the Conclusions chapter.

Suggestions for future research

It would be interesting if the authors could repeat the research, now, after the Covid crisis. I think that a comparative study in this sense would highlight the effects of the pandemic on the purchasing and consumption behavior of fruits and vegetables for families with low incomes who have young children to care for.

Author Response

Thank you for the reviewer comments. We have made the following corrections to the manuscript:

  1. We have strengthened the introduction sections and reviewed the citations throughout the manuscript for recency and relevance.
  2. We have added a conclusions section per the reviewer recommendations (see section 5). 
  3. We have added the suggestion for future research to assess the impact of the COVID-19 pandemic on shopping and consumption behaviors in the conclusions section. 
  4. We have reviewed the entire manuscript for clarity, grammar and coherency. 

Reviewer 2 Report

Dear authors, 

I congratulate you on your research. All the results are very correctly presented and I actually have no comments. Except for a few:

I miss in the methodology a description of the study group - size, gender, age, etc. A description of how the group was recruited. 

Other than that, I know of no major comments

Greetings

Reviewer

Author Response

Thank you for the comments. We have added information regarding the study recruitment, sample size, response rate and related methodology to sections 2.3 and 2.4. We have also reviewed the manuscript further for clarity and coherency. 

Round 2

Reviewer 1 Report

The authors revised the article according to the suggested recommendations.

Congratulations for the article.